# Medium-Term Hydropower Scheduling with Variable Head under Inflow, Energy and Reserve Capacity Price Uncertainty

**Martin N. Hjelmeland [1],\* , Arild Helseth [2] and Magnus Korpås [1]**

[1] Department of Electric Power Engineering, NTNU Norwegian University of Science and Technology, 7491 Trondheim, Norway; Magnus.korpas@ntnu.no

[2] SINTEF Energy Research, 7034 Trondheim, Norway; arild.helseht@sintef.no

\* Correspondence: martin.hjelmeland@ntnu.no

**Abstract:** We propose a model for medium-term hydropower scheduling (MTHS) with variable head and uncertainty in inflow, reserve capacity, and energy price. With an increase of intermittent energy sources in the generation mix, it is expected that a flexible hydropower producer can obtain added profits by participating in markets other than just the energy market. To capture this added potential, the hydropower system should be modeled with a higher level of detail. In this context, we apply an algorithm based on stochastic dual dynamic programming (SDDP) to solve the nonconvex MTHS problem and show that the use of strengthened Benders (SB) cuts to represent the expected future profit (EFP) function provides accurate scheduling results for slightly nonconvex problems. A method to visualize the EFP function in a dynamic programming setting is provided, serving as a useful tool for a priori inspection of the EFP shape and its nonconvexity.

**Keywords:** hydropower scheduling; stochastic programming; integer programming

## 1. Introduction

Increasing rates of renewable energy generation are resulting in a higher demand for flexible power units to balance the power system and to deliver ramping capacity. Regulated hydropower is a flexible renewable energy source that is well suited to provide such services. The increased demand for flexibility has led to the requirement of more detailed optimization models, such that the flexible power units can perform an optimal allocation of resources in the different power markets for energy and ancillary services. In this work, we focus on rotating reserve capacity, providing what is normally referred to as primary and secondary reserves.

The stochastic dual dynamic programming (SDDP) algorithm proposed in [1] is commonly used for hydropower scheduling and can be seen as a sampling-based approach of the nested Benders decomposition proposed in [2]. The sampling-based method for solving multistage stochastic programming problems consists of two main procedures: the forward and backward pass. Instead of visiting all nodes in the scenario tree, the forward pass samples a set of scenarios used to generate candidate solutions. The backward pass follows the trajectories of the candidate solutions computed in the forward pass, starting from the final stage, to approximate the expected future profit (EFP) function. Subsequently, a statistical confidence interval can be computed for controlling the convergence of the method. A more in-depth explanation of the method can be found in [3–5].

In this work, we investigate how improvements in the SDDP algorithm, derived from the stochastic dual dynamic integer programming (SDDiP) algorithm [6], can be used to improve the medium-term hydropower scheduling (MTHS) problem under uncertainty. The MTHS problem normally covers a planning horizon of one to three years, aiming at maximizing a single producer's

expected profit. We have from ongoing research experienced that SDDiP requires considerably more computational force than SDDP [7]. Nevertheless, an improved type of the Benders (B) cuts, called strengthened Benders (SB) cuts, derived from the SDDiP method, show promising results by improving the convergence of the algorithm with a reasonable increase in computation time. The generation of SB cuts requires solving an additional mixed integer programming (MIP) problem to compute the right-hand-side parameter of the cut. This provides an at least as good a cut as the original B cut.

A considerable amount of research has been conducted for solving the nonconvex MTHS problem, such as [8–11]. Except [10], which proposed a novel approach that uses step functions to model a nonconvex EFP function, they all rely on solving some relaxation of the original problem. This is also the case for the SB cuts applied in this work. However, instead of solving the Lagrangian problem to convergence to obtain the cuts, one solves the Lagrangian problem only once, as elaborated in Section 3.2.

For the MTHS problem, a hybrid stochastic dynamic programming (SDP)-SDDP method is currently the state-of-the-art in the Nordic power system. This method was developed in the late 1990s and uses a discrete Markov chain to describe the price uncertainty and an autoregressive model to describe the inflow to the reservoirs [5,12]. The discrete Markov chain is used to circumvent the nonconvexity caused by the bilinear term where the energy price and the generation are multiplied. As the uncertainty is described by two different stochastic processes, it is challenging to model correlations between these processes. For an MTHS problem with weekly decision stages, it has been shown that the correlation between inflow and energy price has not been too significant on a weekly basis, yielding sufficient results by the hybrid SDP-SDDP method [12]. Nonetheless, adding additional markets would extend the dimension of the Markov chain. This comes at a significant increase of computational cost, as presented in [13], where a reserve capacity market was added to the Markov process for an MTHS problem. The recent works [14,15] proposed an elegant approach for including uncertainty in the objective function for dynamic programming (DP) problems, utilizing the fact that the EFP function is a saddle function that is convex w.r.t. to the objective coefficient and concave w.r.t. the state variables. In the following work, we build on the work in [15] to model the MTHS problem with uncertainty of inflow, energy, and reserve capacity price.

*Contributions*

The work carried out in this paper is based on earlier work on developing improved methods to solve the MTHS problem, as in [7,16]. The main contributions are:

- A procedure to visualize and evaluate the shape of the EFP function to give a better insight into the nonconvexities in dynamic programming problems.
- The application of SDDP with SB cuts on a realistic nonconvex MTHS case study. The SDDP model considers correlated stochastic processes of inflow, energy, and reserve capacity price.
- The representation of nonconcave generation functions that are dependent on discharge and water head by concave regions.

## 2. The Medium-Term Hydropower Scheduling Problem

A dense formulation of the MTHS problem is given as the following multistage stochastic programming problem:

$$\max_{(x_1,y_1),\dots,(x_T,y_T)} \mathbb{E}_{\tilde{\xi}}\left\{ \sum_{t=1}^{T} f_t(x_t, y_t, \tilde{\xi}_t) \right\} \tag{1}$$

$$\text{s.t.} \quad Wx_t + Hx_{t-1} + Gy_t = h(\tilde{\xi}_t) \tag{2}$$

$$By_t = 0 \tag{3}$$

$$Cy_t - Dx_t \geq 0 \tag{4}$$

$$Cy_t + Dx_t \leq Cy^{\max} \tag{5}$$

$$(x_t, y_t) \in Y_t \tag{6}$$

$$x_t \in \mathbb{R}^{k_1} \cdot \mathbb{Z}^{k_2}, y_t \in \mathbb{R}^{l_1} \cdot \mathbb{Z}^{l_2} \tag{7}$$

$$\forall t \in \{1, 2, \ldots, T\}. \tag{8}$$

The state variables, $x_t$, carry information between stages with a known initial state, $x_0$. Stage variables are given by $y_t$. The objective is to maximize the expected value of some real value function $f_t$ that describes the profit the system can obtain. The expectation is taken over $\tilde{\xi}_t$, which describes a stochastic process of the inflow to the reservoirs as well as energy and reserve capacity prices. The matrices $W, H, G, B, C, D$ are of suitable dimensions and define the parameters for a given hydropower system. The time-linking constraints in Equation (2) constrain the unit commitment of hydro stations and provide reservoir balances, where the function $h(\tilde{\xi}_t)$ describes the inflows to the reservoirs. The energy balance is given by Equation (3). The system's ability to provide reserve capacity is included in Equations (4) and (5). The generation function defining the relationship between power output, discharge and net head for each station, and the head function, describing how the head is related to reservoir volume, are also described by these constraints. More details on the these functions are given in Section 2.1. Other system constraints, not imperative for this study, are given in Equation (6).

In contrast to earlier characterizations of the MTHS problem, such as [5], where the objective was to maximize income from selling only energy, we extend this to include sales of reserve capacity. In order to keep the MTHS tractable, we define the reserve capacity as a composition of the provision of primary and secondary reserves. Further operational details associated with participation in the different markets are left for the short-term hydropower scheduling (STHS) problem [17] to handle.

## 2.1. Generation Function

The generation function describes a power station's power output. An illustration is given in Figure 1, where the generation is a function of discharge and net head. Since the generation function describes the station's overall power output, one must assume that the units are started by a given sequel. The station consists of two units, as can be seen from the two concave regions along the water discharge in Figure 1.

Due to the computational complexity of the MTHS problem, the generation function is normally cast as a concave function, where the power output only depends on discharge. For a well-regulated and loosely-constrained hydropower system where the producer only considers sales of energy, this assumption is reasonable. Roughly speaking, the optimal solution is to discharge as much water as possible in the hours with the highest energy prices and produce nothing the rest of the year. However, as discussed in Section 1, we expect that hydropower plants will more frequently run at low-level power output in the future to provide ancillary services. Similar behavior might also be imposed by environmental constraints, such as minimum discharge limits for certain periods of the year. While operating at low power outputs, the linear optimization model will observe a higher power output than what is physically feasible and thus overestimate the system's potential profit, as discussed in [7,16]. This overestimation can be avoided by more accurate modeling of the generation function. However, such improvements bring about increased model complexity and computation time. In the following, we define the generation function as a nonconcave function of net head and discharge. The generation function of a power station, for a given stage $t$, is defined by the following:

$$p_c \leq \alpha_i q_c + b_i x_c + \beta_i h, \forall i \in \mathcal{K}(c), \forall c \in \mathcal{C} \tag{9}$$

$$Q_c^{\min} x_c \leq q_c \leq Q_c^{\max} x_c, \forall c \in \mathcal{C} \tag{10}$$

$$P_c^{\min} x_c \leq p_c \leq P_c^{\max} x_c, \forall c \in \mathcal{C} \tag{11}$$

$$\sum_{c \in \mathcal{C}} x_c \leq 1 \tag{12}$$

$$x_c \in \{0,1\}, (p_c, q_c) \in \mathbb{R}_+ \tag{13}$$

where $h$ is the net head, i.e., height difference between the station's upstream reservoir and the tailwater level. The set $\mathcal{C}$ contains concave regions of the generation function, where the discharge, $q_c$, and generation, $p_c$, are constructed for each of these regions. For each concave region, the generation function is bounded above by a set of hyperplanes, $\mathcal{K}(c)$, with coefficients $\alpha_{i(c)}$ and $\beta_{i(c)}$, and a right-hand side parameter $b_{i(c)}$. The generation function describes the entire station's power output; therefore, one must make an assumption that the units are started in a certain sequence. This sequence of starting up new units leads to nonconcavities in the generation function, as seen in Figure 1. Furthermore, another source of nonconcavity comes from the fact that power output for a hydropower station is a nonconcave function w.r.t. head. Therefore, one must make a trade-off between accurate problem formulation and computation time. To tackle this, we base our implementation on the generation function presented in [16].

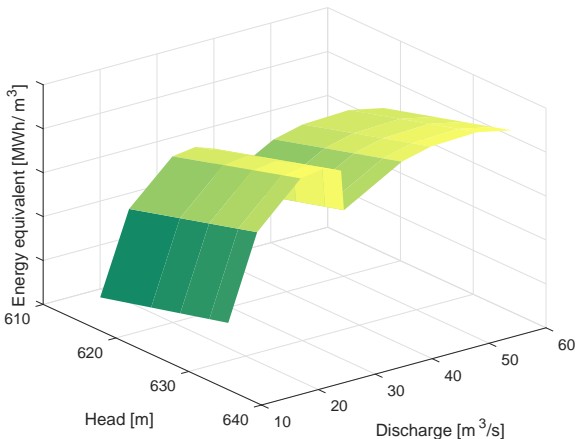

**Figure 1.** A generation function dependent on net head and discharge. For the purposes of illustration, the z-axis is given by the energy equivalent [MWh/m$^3$].

Several authors have investigated how to include variable head in the hydropower scheduling problem. Most of the publications are related to solving the STHS problem; see [17–20]. There has also been some work conducted on how to include variable head in MTHS problems [8,21]. A nonconcave generation function is described in [8], using a piecewise-linear approximation. The formulation in [8] requires one binary variable for each discrete point of the generation function, compared to one binary variable for each concave region in the generation function, as in Equations (9)–(13). In [21], the generation function is approximated by hyperplanes, and the authors propose a quadratic function to describe the relationship between head and reservoir volume. The bilinear terms are divided into a grid with different cells, where each cell is represented by McCormick envelopes [22]. Our approach avoids the bilinear term as the generation function is described by a set of hyperplanes for each of the concave regions. The method of using hyperplanes to describe the generation function is not novel, e.g., as proposed in [20], so our approach is thus an extension to an already established methodology.

The head function, relating head and reservoir volume, can for most Norwegian reservoirs be approximated by a concave function without significantly compromisingaccuracy. Most Norwegian

hydropower plants have a relatively high head, and generation is typically less dependent on head variations than it is for hydropower systems in other parts of the world. An illustration of the head function is given in Figure 2, where the head function is illustrated with some constructed reservoirs. One can observe that the head function is concave for a reservoir with a monotonically-increasing cross-section, but it may be nonconcave for a reservoir that inhabits a subsurface cave.

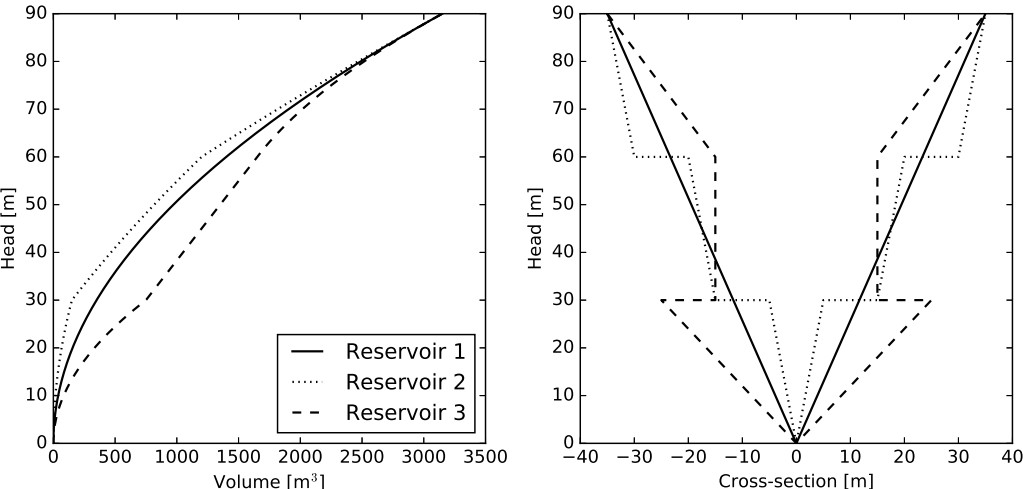

**Figure 2.** (**Right**) Cross-section of some artificial hydropower reservoirs. (**Left**) Head as a function of reservoir volume for the same reservoirs (corresponding line style). It is assumed that, from the view of the cross-section, the reservoirs has a depth of one unit.

### 2.2. Stochastic Processes

To solve the MTHS problem with uncertain inflow, energy, and reserve capacity price, we apply a vector autoregressive model of order one (VAR-1); see Equation (14).

$$n_t = \Phi n_{t-1} + \tilde{\xi}_t \tag{14}$$

$$r_t = \mu_t + \sigma_t n_t. \tag{15}$$

where $n_t$ is the vector of the normalized stochastic processes, $\Phi$ is a time-invariant correlation matrix, and $\tilde{\xi}_t$ is a vector of white noise with realizations denoted as $\tilde{\xi}_t$. The physical realizations $r_t$ of the normalized variables are given by Equation (15), where $\mu_t$ and $\sigma_t$ are the expected value and standard deviation of the processes. $p_t$ is defined as the subset of $r_t$ that describes the objective term coefficients. As described in [15], these coefficients must be computed a priori to solving the stage-wise decision problem. Thus, the energy and reserve capacity prices are found beforehand and provided as parameters to the stage-wise decision problem, while the constraints on the normalized inflows are included in the weekly decision problem. Note that the normalized inflows can be calculated a priori and provided to the optimization problem as a parameter, but for modeling convenience, they are added as constraints. The treatment of objective term uncertainty in the SDDP method was first described in [15] and is further discussed in Section 3.3.

### 3. Methodology

The following section first describes how one can visualize and inspect the EFP function. Following that, it gives some insight into how the uncertain objective function is included and how the SB cuts introduced in [6] are used to solve the MTHS problem with variable head and price uncertainty.

### 3.1. The EFP Visualization Approach

In order to gain insight into the shape of EFP function, we solve the extensive form of the MTHS problem given by Equations (1)–(8). By visually inspecting the EFP function, one can get a first-hand impression whether it can be approximated using SB cuts with sufficient accuracy or not.

To solve the extensive MTHS problem, we rely on a tractable scenario tree representing some of the underlying uncertainty with a one-year planning horizon. Then, by looping over different initial reservoir levels, re-solving the MTHS problem, and storing the objective value, one can obtain an estimate of the EFP function for the first stage. This procedure is performed for some cases of the MTHS problem with and without capacity reserves, as seen in Figure 3.

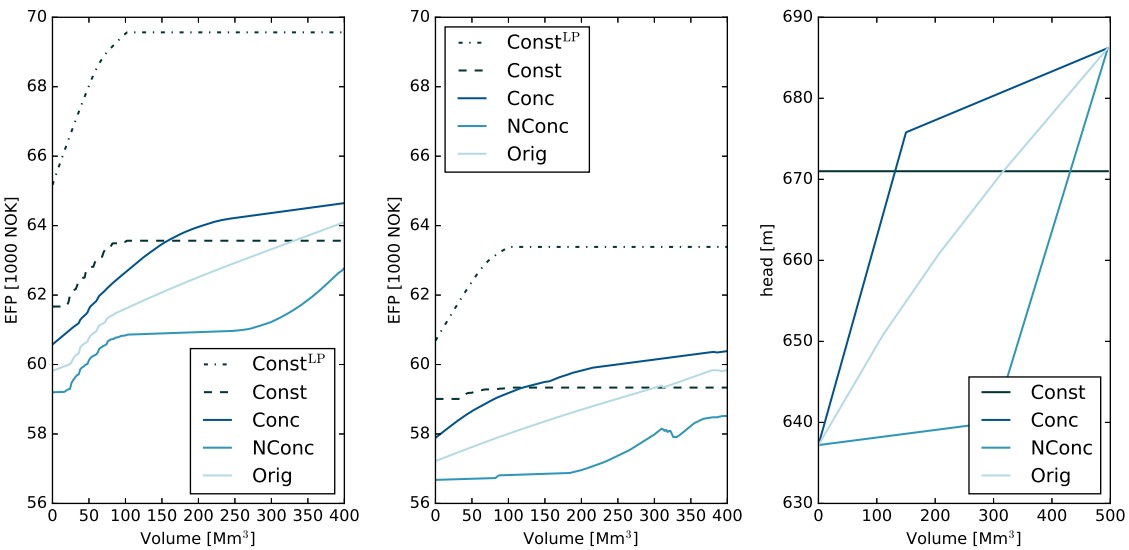

**Figure 3.** (**Left**) EFP with sales of reserve capacity and energy. (**Middle**) EFP with sales of energy. (**Right**) Different head functions. The colors of the lines in the two leftmost figures refer to the head function, $h(V)$, in the right figure. The different head functions are constant (Const), concave (Conc), original (Orig) and nonconcave (NConc), respectively. The Const$^{LP}$ function refers to the linear programming relaxation of the MTHS problem. Both dashed lines refer to the constant head function; thus, the power output only depends on discharge.

The approach is limited in the sense that it only provides an approximation of the EFP function for a given time stage. Moreover, it can only provide meaningful visualizations a few state variables at a time. The results should therefore only assist the modeler in the choice of what approach to use. Note that this visual approach can be combined with numerical indices, such as those presented in [23], to indicate the degree of nonconcavity of the MTHS problem.

Inspecting the EFP for the MTHS at Hand

The MTHS problem, given by Equations (1)–(8), was defined with different head functions; a highly concave, highly nonconcave, constant and the actual function for a given reservoir. They are illustrated in the right plot in Figure 3. The resulting EFP function is given in the two leftmost plots, where the plot to the left includes sales of reserve capacity. It is clear that the shape of the head function has a significant impact on the corresponding shape of the EFP function, yielding information that can be used for deciding which solution strategy, and possible required simplifications, one should use.

The two EFP functions corresponding to a constant head are plotted (in black, dashed lines). One of them is the linear programming (LP) relaxation, denoted as PQH$^{LP}$. As expected, the LP relaxation yielded a concave EFP function. Moreover, the LP relaxation significantly overestimates the EFP function. Since the EFP function with the actual head function is not a highly nonconcave

function, we expect that representing the generation function by piecewise-linear functions (cuts) provides acceptable results.

*3.2. DP Formulation*

In SDDP, the forward pass is used to generate valid candidate solutions that are used for computing the EFP function in the backward pass. All possible candidate solutions generated in the forward pass must, therefore, be present in the solution space in the problem used in the backward pass. The EFP function is described by an upper approximated piecewise-linear concave function, generated in the backward pass. For the forward pass, we define the following DP problem for iteration $i$:

$$\text{FP}_t^i : \quad Q_t^i(x_{t-1}, u_{t-1}, \xi_t) := \max_{x_t, y_t, z_t, u_t} f_t(x_t, y_t, u_t, \xi_t) + \phi_t^i(x_t, u_t, \tilde{\xi}_{t+1}) \tag{16}$$

$$\text{s.t.} \quad (z_t, x_t, y_t) \in X_t(\xi_t) \tag{17}$$

$$z_t = x_{t-1} \tag{18}$$

$$(z_t, u_t) \in \mathbb{R}, (x_t, y_t) \in \mathbb{R} \cdot \mathbb{Z}. \tag{19}$$

The objective function (16) consists of the present profit function, $f_t$, and the EFP function $\phi_t^i$. The problem is constrained by the set $X_t$ and the copy constraint Equation (18) as described in [6] together with the copy variable $z_t$. In addition to the state and stage variables ($x_t$ and $y_t$), the additional state variable $u_t$ is included in the formulation to include the uncertain objective coefficients, as discussed in [15].

In order to compute the EFP function, we define a backward pass problem, $\text{BP}_t^i$, where integrality has been relaxed.

$$\text{BP}_t^i : \quad Q_t^i(x_{t-1}, u_{t-1}, \xi_t) := \max_{x_t, y_t, z_t, u_t} f_t(x_t, y_t, u_t, \xi_t) + \phi_t^i(x_t, u_t, \tilde{\xi}_{t+1}) \tag{20}$$

$$\text{s.t.} \quad (z_t, x_t, y_t) \in X_t(\xi_t) \tag{21}$$

$$z_t = x_{t-1} \tag{22}$$

$$(z_t, u_t, x_t, y_t) \in \mathbb{R}. \tag{23}$$

By solving $\text{BP}_t^i$, we obtain the cut coefficients $\pi_t^i$, from the dual values of the copy constraint in Equation (22). The cut coefficients aligned with the objective-term uncertainty are purely given by the sampled value, as described in [15]. Further, the cut used to describe the EFP function is enhanced by solving the following Lagrangian problem based on a Lagrangian relaxation of problem $\text{FP}_t^i$.

$$\text{LG}_t^i : \quad \mathcal{L}_t^i(\pi_t) := \max_{x_t, y_t, z_t, u_t} f_t(x_t, y_t, u_t, \xi_t) + \phi_t^i(x_t, u_t, \tilde{\xi}_{t+1}) - \pi_t^\top z_t \tag{24}$$

$$\text{s.t.} \quad (z_t, x_t, y_t) \in X_t(\xi_t) \tag{25}$$

$$(z_t, u_t) \in \mathbb{R}, (x_t, y_t) \in \mathbb{R} \cdot \mathbb{Z}, \tag{26}$$

By solving the Lagrangian problem, one can obtain the SB cut, as proposed in [6]. Note that the constant term $\pi_t^\top x_{t-1}$ is neglected in the Lagrangian problem as it would be subtracted in the SB cut, which is given as;

$$\theta_t \leq \sum_{m \in \mathcal{C}(t)} q_{tm} \mathcal{L}_m(\pi_m^i) + \sum_{m \in \mathcal{C}(t)} q_{tm}(\pi_m^i)^\top x_t - (p_t^i)^\top \mu_t. \tag{27}$$

Similarly, the B cut is given as:

$$\theta_t \leq \sum_{m \in \mathcal{C}(t)} q_{tm} \mathcal{Q}_m^{i*} + \sum_{m \in \mathcal{C}(t)} q_{tm}(\pi_m^i)^\top (x_t - x_t^*) - (p_t^i)^\top \mu_t. \tag{28}$$

where $\mathcal{Q}_m^{i*}$ is the objective value of $\text{BP}_m^i$ and $x_t^*$ is the candidate solution. The problem is solved for $m \in \mathcal{C}(t)$, where $\mathcal{C}(t)$ is the set of children nodes from a node in stage $t$ and $q_{tm}$ is the conditional probability. Recall that SDDP requires the stochastic variables to be stage-wise independent; therefore, the set $\mathcal{C}(t)$ is the same for all nodes in time stage $t$. The function $\phi_t^i(x_t, u_t, \tilde{\xi}_{t+1})$ is thus confined by some upper bound and the acquired set of B or SB cuts.

### 3.3. Uncertainty Modeling

In the following, we provide some insight into how the objective term uncertainty modeling is done. For the purposes of illustration, we assume that all state variables are fixed and that we only look at the terms where the auxiliary term $u_t$ is present. Assume that two samples of the objective term coefficient $p_t^1$ and $p_t^2$ are available and that two cuts were constructed around these. Subsequently, a third sampling is done, and the problem to be solved can be given as:

$$\max_{u_t} \left\{ p_t^3 u_t + \theta_t : \theta_t \leq C_t^i - p_t^i u_t, (\theta_t, u_t) \in \mathbb{R}_+, \forall i \in \{1,2\} \right\}. \tag{29}$$

Since the state variables are assumed fixed, they are embedded in the constant term $C_t^i$. One can see that the problem consists of maximizing the present profit, $p_t^3 u_t$, and future profit, described by the two cuts. The problem given by Equation (29) is, therefore, able to assert whether there is an expectancy for greater profits in the future or not, depending on the current realization of the objective term coefficient. An illustration of this is given in Figure 4.

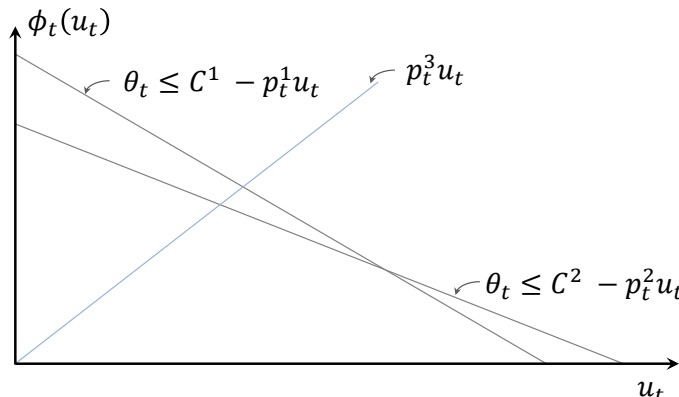

**Figure 4.** Illustration of the representation of the uncertain objective term price. The EFP function w.r.t. to the auxiliary variable $u_t$ is given, and $x_t$ is assumed fixed and added to the parameter $C$. Observe that the cuts represent the potential for future profit, whereas the sampled objective term coefficient $p_t^3$ describes the present profit potential. Thus, the model can compute the trade-off between them.

The algorithm for how the MTHS problem is solved is given in the next section.

### 3.4. Solution Approach

We define Algorithm 1 based on the SDDP framework. As seen from Lines 14 and 16 in the algorithm, one can choose which type of cut (B or SB) to use. After a certain amount of iterations, a final simulation is carried out on a larger set of scenarios.

Note that convergence cannot be guaranteed as $\text{FP}_t^i$ is a nonconcave function w.r.t. the state variables. However, the approach will give an approximate solution that can yield good results depending on how nonconcave the true EFP function is. The nonconvexity of the EFP function can be visualized by the approach proposed in Section 3.1. Other measures to characterize how prominent the nonconcavities are could also be performed, as in Chp. 7.2. of [23].

---

**Algorithm 1:** Solving the MTHS problem.

---

1  Set $x_0^i$, $i \leftarrow 1$, UB $= +\infty$, and LB $= -\infty$

2  **while** $i < i^{\max}$ **or some other stopping criteria do**

3      Sample $N$ scenarios $\Omega^i = \zeta_1^k, \ldots, \zeta_{Tk=1,\ldots,N}^k$

    /* Forward iteration                                                                                                 */

4      **for** $k = 1,\ldots,N$ **do**

5          **for** $t = 1,\ldots,T$ **do**

6              Solve $\text{FP}_t^i$, and collect solution $f_t$ from Equation (16)

7          $\text{lb}^k \leftarrow \Sigma_{t=1,\ldots,T} f_t$

    /* Compute lower bound                                                                                                */

8      $\mu \leftarrow \frac{1}{N}\Sigma_{k=1}^N \text{lb}^k$ and $\sigma^2 \leftarrow \frac{1}{N-1}\Sigma_{k=1}^N (\text{lb}^k - \mu)^2$

9      LB $\leftarrow \mu + z_\alpha \frac{\sigma}{\sqrt{N}}$

    /* Backward iteration                                                                                                */

10      **for** $t = T, \ldots, 2$ **do**

11          **for** $k = 1,\ldots, N$ **do**

12              **for** $m \in \mathcal{C}(t)$ **do**

13                  Solve $\text{BP}_t^i$, and collect $\pi_m^i$ from Equation (22)

14                  **if** *B cuts* **then**

15                      Collect $\mathcal{Q}_t^i$ from Equation (20)

16                  **else if** *SB cuts* **then**

17                      Solve $\text{LG}_t^i$, and collect $\mathcal{L}_t^i$ from Equation (24)

18              Collect $p_t \subset r_t$ from Equation (15)

19              Add desired cut to $\phi_t^i$

    /* Compute upper bound                                                                                                */

20      UB $\leftarrow Q_1^i(x_0^i, u_0, \zeta_0^i)$

21      $i \leftarrow i + 1$

/* Final simulation                                                                                                       */

22  Sample $M$ scenarios $\Omega^i = \zeta_1^k, \ldots, \zeta_{Tk=1,\ldots,M}^k$

23  **for** $k = 1,\ldots,M$ **do**

24      **for** $t = 1,\ldots,T$ **do**

25          Solve $\text{FP}_t^i$, and collect solution $f_t, x_t^{ik}, y_t^{ik}$ from Equation (16)-(19)

26      $\text{lb}^k \leftarrow \Sigma_{t=1,\ldots,T} f_t$

/* Compute lower bound                                                                                                    */

27  $\mu \leftarrow \frac{1}{M}\Sigma_{k=1}^M \text{lb}^k$ and $\sigma^2 \leftarrow \frac{1}{M-1}\Sigma_{k=1}^M (\text{lb}^k - \mu)^2$

28  LB $\leftarrow \mu + z_\alpha \frac{\sigma}{\sqrt{M}}$

---

## 4. Case Study

The case study is a representation of a Norwegian hydropower system, comprised of three reservoirs and two power stations. The power stations have 13.8 MW and 365 MW of installed capacity. There is a short-term and long-term reservoir connected to the largest power station, as seen in Figure 5. Thus, the aim for the long-term reservoir is to store as much water for usage in the most remunerated hours during the year, while the short-term reservoirs need to be properly managed in order to avoid spillage. This system was also used as a study case in [7,24].

The VAR-1 model representing the stochastic processes was fitted to 70 historical years of inflow and energy prices obtained from a fundamental market model generating energy prices for those 70 years [25]. Historical prices for the primary reserve market are used to describe the reserve capacity prices.

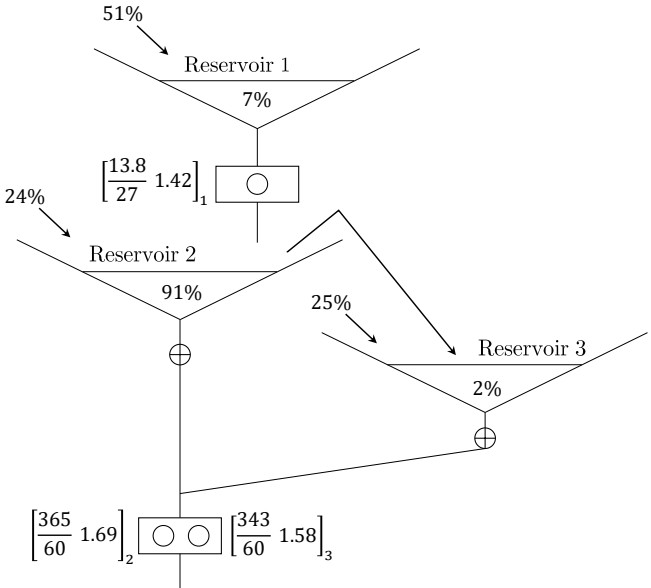

**Figure 5.** Illustration of the hydropower system. There are three reservoirs, each represented by its relative storage capacity and inflow compared to the system as a whole. As an example, Reservoir 1 has 7% of the system's storage capacity and 51% of the inflow. Reservoirs 2 and 3 have a hatch downstream that controls which reservoir is depleted. Only one of the reservoirs can be depleted at a time, due to their different head. Reservoir 2 can also bypass water, indicated by the arrow between the reservoirs. The hydropower stations are represented by their maximum power at nominal head, discharge and the energy equivalent ($\frac{MW}{m^3/s}$ MWh/m$^3$). Since the lower power station is connected to two reservoirs, it has different efficiencies, depending on which reservoir is depleted.

We use weekly decision stages. Each weekly decision problem consists of 1858 constraints (not considering the cuts) and 1152 variables (837 continuous and 315 binary). There are 104 weeks in the scheduling horizon, and we consider 15 branches in the backward pass of the SDDP algorithm. Each week has 21 time-steps representing three time blocks of the day. Three scenarios are sampled for each forward SDDP iteration, and the final simulation is carried out with 300 scenarios. We use the same sampled scenarios for all cases. From the final simulation, a confidence interval is computed. The problem was formulated in C++ with Gurobi 7.5 as the optimization solver. The computations were carried out on a computer cluster with two Intel Xeon E5-2690 v4 processors, 2.6 GHz, and 384 GB RAM. No parallelization was carried out except the one from the optimization solver. Parallelization in the SDDP framework is well studied, as in [26], and thus neglected in this work. It would, however, contribute to significantly reducing the CPU time.

*Results and Discussion*

In the following, the results from Algorithm 1 are outlined with the use of both B and SB cuts. The MTHS problem was first solved with uncertainty of inflow, reserve capacity, and energy price. A case with only uncertainty in inflow was performed for comparison.

The convergence plot for Algorithm 1 is shown in Figure 6, when B and SB cuts were used. It is clear that the SB cuts provided significantly tighter cuts and a better policy. One can also observe that the upper bound converged slowly and would most likely continue to improve with more iterations. The computation time did, however, become more prominent for SB cuts, as seen in Table 1. Algorithm 1 required approximately five-times more time with SB cuts than with B cuts. The use of parallel processing could easily drive the computation time down, making the use of SB cuts better suited for daily operational use.

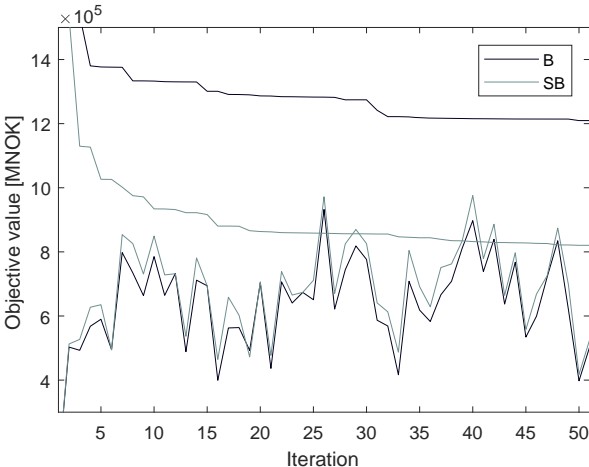

**Figure 6.** Convergence of the approach with Benders and strengthened Benders cuts.

**Table 1.** Economic and computational performance of the two case studies with use of either B or SB cuts. Uncertainty of inflow, reserve capacity, and energy price (top) and uncertainty of inflow (bottom).

|    | UB (kNOK) | stat. LB (kNOK) | Gap (%) | Time (h min) |
|----|-----------|-----------------|---------|--------------|
| B  | 1,209,510 | 502,838         | 58.4    | 5 17         |
| SB | 820,466   | 525,493         | 36.0    | 23 59        |
| B  | 1,017,580 | 537,808         | 47.1    | 5 52         |
| SB | 614,616   | 563,280         | 8.4     | 34 31        |

The expected value of water, or water values (WV), for the largest reservoir in the system are shown in Figure 7. The WVs were computed by fixing all the other state variables in the EFP function and finding the coefficients of the binding cuts. One can observe that even though the cut coefficients from both B and SB cuts came from the problem $BP_t^i$, the water values for the SB cuts were generally lower than the B cuts. This can be seen as a result of the right-hand-side in the SB cuts being lower, therefore lowering the cuts, which resulted in a lower water value for the same state, as the EFP function is concave. An illustration of this is given in Figure 8, where two cuts are generated in the first iteration of the SDDP algorithm. The coefficients of the cuts were the same for both SB and B cuts, but as seen from the top left and right plot, the SB cuts had a tighter right-hand side. This results in the water values given in the lower plot. After consecutive iterations, the water values with SB cuts tended to stabilize on lower values, which is a reasonable observation as the SB cuts made the model see less expected profit in the future.

A percentile plot of the reservoir trajectories for the largest reservoir is shown in Figure 9, for both B and SB cuts. The figure clearly shows how the algorithm was able to utilize the reservoir better when SB cuts were used. When B cuts were used, the algorithm saw a higher expected future profit than what was achievable, resulting in a simulated operation at very high reservoir volumes. Implications of this can be seen in Figure 7b,d, where the change in water values is substantial around Week 18, due to the spring flood, giving a high risk of spillage for large reservoir volumes.

In Table 1, the bounds of the algorithm are shown, computed from the final simulation, together with the percentage-wise gap and computation time for the 50 forward and backward iterations. For comparison, the problem was solved with only uncertainty of inflow, which is given in the bottom half of the table. Observe how the convergence properties improve, indicating that the approach of including objective term uncertainty in SDDP by [15] requires more iterations. In validation studies using a smaller system with fewer decision stages, it was observed that the approach does slowly converge. This illustrates the difficulty of solving multi-stage stochastic problems with high dimensions of uncertainty.

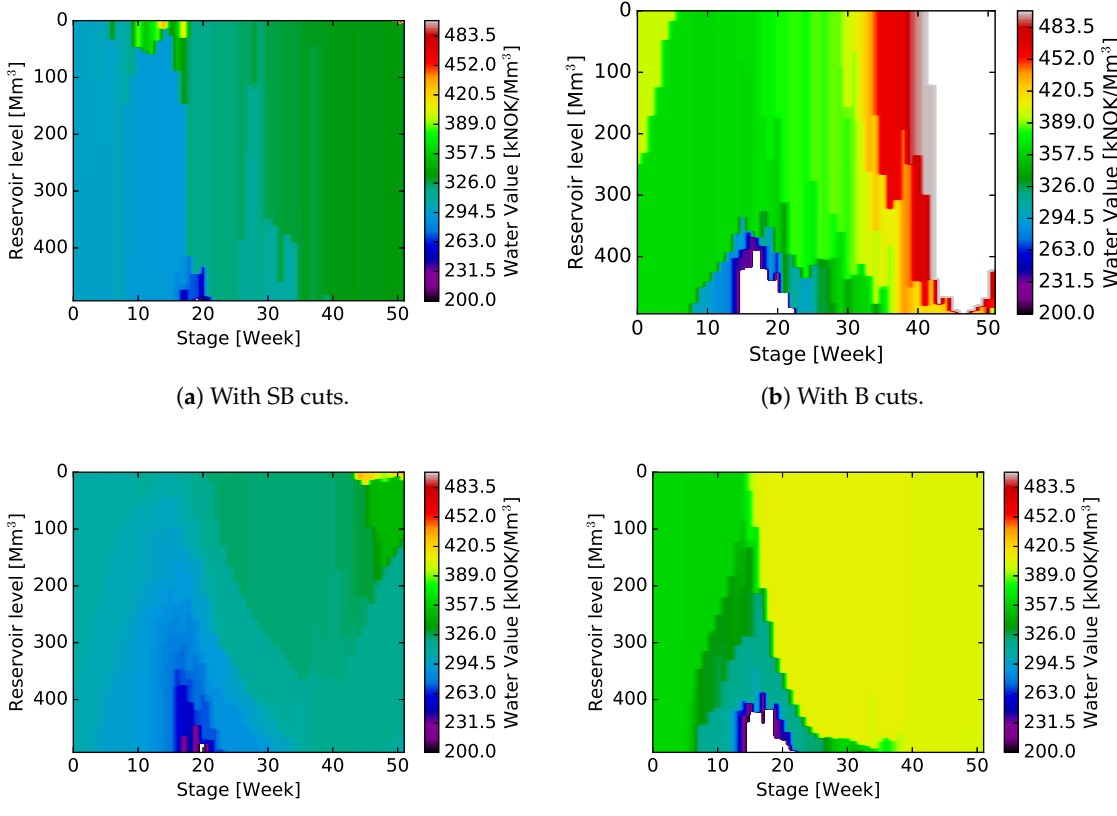

(**a**) With SB cuts.

(**b**) With B cuts.

(**c**) With SB cuts and only uncertainty of inflow.

(**d**) With B cuts and only uncertainty of inflow.

**Figure 7.** Water values for Reservoir 2 for the different case studies.

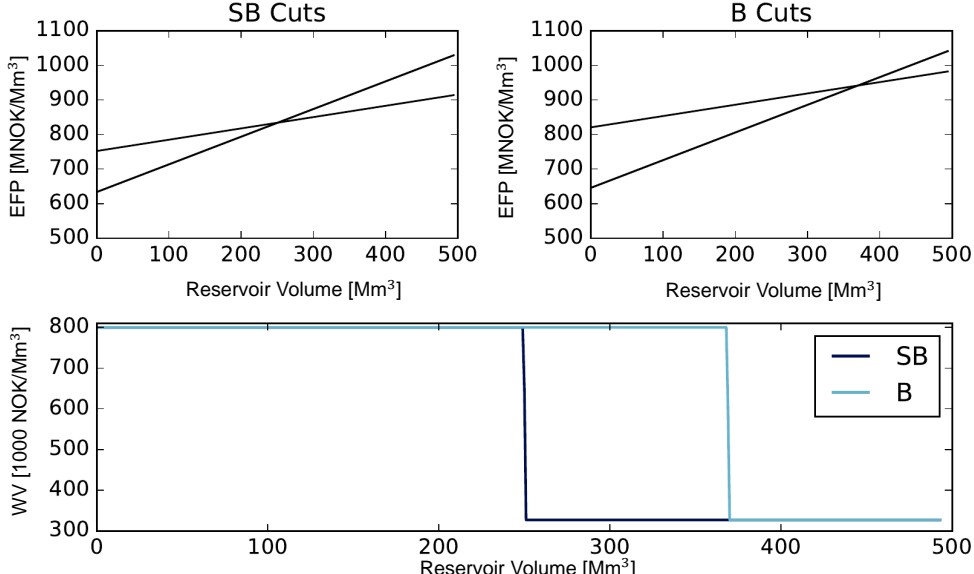

**Figure 8.** Illustration of the B and SB cuts and how this affects the water value.

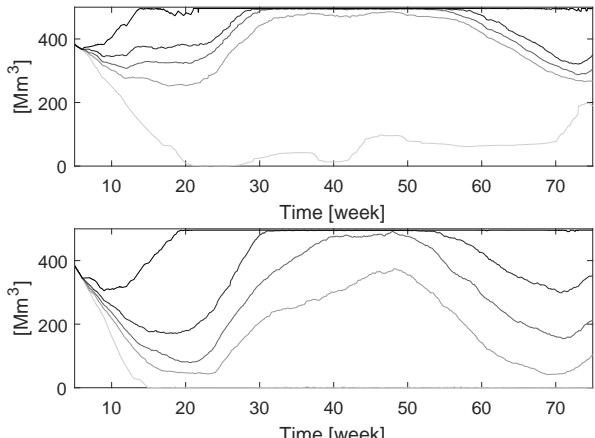

**Figure 9.** Percentile plot of reservoir trajectories for Reservoir 2. (**Top**) With Benders cuts and (**bottom**) with strengthened Benders cuts.

## 5. Conclusions

A medium-term hydropower scheduling (MTHS) problem with variable head and uncertainty in inflow, reserve capacity, and energy price was investigated. The proposed model based on the stochastic dual dynamic (SDDP) method included correlations between the different stochastic processes and allowed for representation of a detailed hydropower system.

By means of visualization, we found that the expected future profit (EFP) function for the MTHS case study was not highly nonconcave, and we argue that the approximation of the EFP as a concave function within the SDDP method is a fair compromise between accuracy and computation time. We compared two types of Benders cuts to approximate the EFP function, namely the Benders (B) and the strengthened Benders (SB) cuts.

In the presented case study, it was found that the use of SB cuts provided a significantly better policy than with the use of B cuts. The policy improvement comes at an increased computational time, around five-times higher for SB than B in the case study. Moreover, we found that the inclusion of objective term uncertainty led to significantly slower convergence.

**Author Contributions:** Conceptualization, M.N.H., A.H., and M.K.; methodology, M.N.H. and A.H.; software, M.N.H.; validation, M.N.H., A.H., and M.K.; writing, original draft preparation, M.N.H.; writing, review and editing, A.H. and M.K.; visualization, M.N.H.

**Funding:** This research was funded by the Research Council of Norway and industry partners under Grant 228731/E20.

**Conflicts of Interest:** The authors declare no conflict of interest.

## Abbreviations

The following abbreviations are used in this manuscript:

| | |
|---|---|
| B | Benders |
| DP | Dynamic programming |
| EFP | Expected future profit |
| LP | Linear programming |
| MIP | Mixed integer programming |
| MTHS | Medium-term hydropower scheduling |
| SB | Strengthened Benders |
| SDDiP | Stochastic dual dynamic integer programming |
| SDDP | Stochastic dual dynamic programming |
| SDP | Stochastic dynamic programming |
| WV | Water values |

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
