# Peer review of "Medium-Term Hydropower Scheduling with Variable Head under Inflow, Energy and Reserve Capacity Price Uncertainty"

_energies, doi:10.3390/en12010189_

Round 1

Reviewer 1 Report

This study reports a very complete model for hydropower scheduling which accounts for a wide range of variables. The paper is acceptable for publication after addressing the following points:

Line 131: is this approximation valid for reservoirs in other areas different than Norway? How would it impact the results?

Line 55: repetition of “in”.

Section 5.3: it would be interested to include/refer to other uncertainty quantification methods, such as techniques with sensitivities/Sobolev indices as in Rezaeiravesh et al. Eur. J. Mech. B/Fluids 72, 2018.

Line 247: add comments on how to parallelize this algorithm.

Figure 8: consider presenting this figure with contours instead of 3D surfaces to ease its visualization.

It would be interesting to add some comments regarding the general applicability of these methods to other variable energy sources such as concentrated solar power (CSP), see for instance Cachafeiro et al. Energy Procedia 69, 2015.

Author Response

See attachement.

Reviewer 2 Report

1) MTHS becomes more and more important and complex, the article is a significant contribution in the field. It has a solid base in earlier and ongoing research, the model case is highly relevant. It is well structured and written, I have only minor comments.

2) SDDP and B/SB cuts are really not within my competence but as far as I can follow the presentation, it makes a convincing impression.

3) In figure 5 the reservoires should be numbered. In figure 8, there are different scales in a and b which makes a comparison difficult. Regardless of this, the statement on rows 262 - 264 is not convincing (i e it is difficult to see in figure 8 the ground for the statement). In figure 8 B is to the left, in figure 9 just below B is to the right, change that.

4) Some small spelling errors on rows 35, 126, 137, figure 8d, 348.

Reviewer 3 Report

The reviewer wants to thank the authors for their very inserting paper. He/she has only some small suggestions/comments/questions: 

-      Line (L) 20: proposed in Pereira [1] … please always use the author(s) name and not only the []. Similar in L22,28,40 and so on. Please check the whole paper.

-      L 78: Please use Equation or Eq.~(1b). It is obvious based on the () that an equation is referenced, but it always stopped me in reading. Please check the whole paper. 

-      L91: the two turbines are equal? Otherwise there should be another set of generation function (starting with the first and one with the other turbine)?  

-      L100-101: Not only minimum discharges are relevant but also the possibly to store water in case of the danger of flooding the downstream area. The following sentences may be a little bit misleading. After the first reading the reviewer thought, that the authors would include such limitation in the generation function. Please include the flooding events and clarify this in the text. 

-      Fig. 5 / Case study: The water out of the reservoir 1 with hydro power plant (HPP) 1 flows only into the reservoir 2 or is it split into R2 and 3…. The reviewer doesn’t understand the arrow from 2 to 3. Is this a bypass? Furthermore, R2 and R3 seem to have different heights but in reality, those levels are connected based on the fact that the penstocks are joint together? Please clarify this. 

The paper is very well written and the reviewer is looking forward to read the paper again.

Thank you!

Round 2

Reviewer 1 Report

The revised version addresses most of the concerns and constitutes an improved manuscript. I still believe that adding the reference to Cachafeiro et al., highlighting the possibility of using this method in varying energy resources like CSP, would improve the general applicability of the article.

Author Response

Thank you for the comments.

Reviewer 2 Report

My comments are all considered by the authors, the conclusion is that the manuscripts now warrants publication in Energies.

Author Response

Thank you.

Reviewer 3 Report

The reviewer wants to thank the authors again and yes, there is no explicate mentioning of the author(s) name inclusion for citations in the text. Nevertheless, it is commonly used and better to read. This is the reviewer’s opinion and doesn’t change the quality of the paper. 

But there is one point, which should be clarified: Figure 5 only shows a valve after the connection of the reservoir 3. This would not stop the water levelling out between those two reservoirs. The presented hydraulic sketch is misleading and has to be corrected. Please change Figure 5 and clarify this not only in the caption. 

Thank you.

Author Response

Thank you for valuable comments and pointing out the error in Figure 5. We have fixed this is in the latest draft.